# Usability Study of a Multicomponent Exergame Training for Older Adults with Mobility Limitations

**DOI:** 10.3390/ijerph182413422

**Published:** 2021-12-20

**Authors:** Melanie Thalmann, Lisa Ringli, Manuela Adcock, Nathalie Swinnen, Jacqueline de Jong, Chantal Dumoulin, Vânia Guimarães, Eling D. de Bruin

**Affiliations:** 1Department of Health Sciences and Technology, Institute of Human Movement Science and Sport, ETH Zürich, 8093 Zurich, Switzerland; m.thalmann@ggaweb.ch (M.T.); lisa@ringli.ch (L.R.); manuela@dividat.ch (M.A.); 2Department of Rehabilitation Sciences, KU Leuven, 3001 Leuven, Belgium; nathalie.swinnen@kuleuven.be; 3University Psychiatric Centre, KU Leuven, 3070 Kortenberg, Belgium; 4Physio SPArtos, 3800 Interlaken, Switzerland; j.dejong@artos.ch; 5School of Rehabilitation, Faculty of Medicine, Université de Montréal, Montreal, QC H3T 1J4, Canada; chantal.dumoulin@umontreal.ca; 6Fraunhofer Portugal AICOS, 4200-135 Porto, Portugal; vania.guimaraes@fraunhofer.pt; 7Division of Physiotherapy, Department of Neurobiology, Care Sciences and Society, Karolinska Institute, 171 77 Stockholm, Sweden; 8Department of Health, OST—Eastern Swiss University of Applied Sciences, 9001 St. Gallen, Switzerland

**Keywords:** exergame, mobility limitations, usability, fall prevention, geriatric giants, healthy ageing

## Abstract

The global population aged 60 years and over rises due to increasing life expectancy. More older adults suffer from “geriatric giants”. Mobility limitations, including immobility and instability, are usually accompanied by physical and cognitive decline, and can be further associated with gait changes. Improvements in physical and cognitive functions can be achieved with virtual reality exergame environments. This study investigated the usability of the newly developed VITAAL exergame in mobility-impaired older adults aged 60 years and older. Usability was evaluated with a mixed-methods approach including a usability protocol, the System Usability Scale, and a guideline-based interview. Thirteen participants (9 female, 80.5 ± 4.9 years, range: 71–89) tested the exergame and completed the measurement. The System Usability Scale was rated in a marginal acceptability range (58.3 ± 16.5, range: 30–85). The usability protocol and the guideline-based interview revealed general positive usability. The VITAAL exergame prototype received positive feedback and can be considered usable by older adults with mobility limitations. However, minor improvements to the system in terms of design, instructions, and technical aspects should be taken into account. The results warrant testing of the feasibility of the adapted multicomponent VITAAL exergame, and its effects on physical and cognitive functions, in comparison with conventional training, should be studied.

## 1. Introduction

The global population aged 60 years and over is rising due to increasing life expectancy. The World Health Organization forecasts that the number of older people over 60 years will double from 1 billion in 2020 to almost 2.1 billion in 2050 [1]. Age- and behavior-related diseases and disabilities accompany the ageing population. Consequently, more older adults suffer from one of the “geriatric giants” defined by Bernhard Isaacs as immobility, instability, incontinence, and impaired intellect [2]. With the growing older population, there has been major interest in preventing age-related problems that cause morbidity and mortality and in maintaining and improving the quality of life of older adults.

Mobility is by far one of the most important factors affecting independence and quality of life in older adults [3,4]. A gap between an individual’s abilities and environmental demands can be defined as mobility limitations or disabilities [4], such as immobility and instability. Different studies [5,6,7,8] report that 23 to 47% of older adults have mobility limitations. Mobility limitations are potentially associated with a higher risk of falls [3,9,10,11], disability [12], mortality [13,14], and worsening of psychosocial health due to social isolation [3,15,16]. Such limitations not only dramatically affect the lives of those living with the condition, but often confer a severe burden on families, friends, caregivers [17], and healthcare systems [18]. Mobility limitations are usually accompanied by physical and cognitive decline and can be further associated with gait changes, which might be the reason for the increased risk of falling [3,5,19,20,21,22].

Regular physical activity in older adults positively affects health state, gait speed, and stability, as well as general well-being [23]. Exercise interventions, which aim to improve physical functions such as strength or balance, have been shown to reduce fall rates and risks [24,25,26]. However, not only age-related declines in physical functions are responsible for gait impairments and higher risks of falls but also reduced cognitive functions, such as attention and executive functions [27,28,29,30,31]. Additionally, a decline in cognitive functions has been associated with an increased incidence of mobility limitations [32] as, for all movements and control of physical functions (besides some reflexes), the brain and cognitive functions are involved. Consequently, combined cognitive–motor training is required for the most effective prevention of mobility limitations and falls [33,34,35]. Individualized exercise interventions may slow down disability progression in older adults before it impacts quality of life [36,37].

Considering that rather long-term exercise is required for maintaining functional capacity [38], preventive exercise measures should ideally become an integral part of daily life and be easily accessible. No access to public health centers and training facilities (e.g., due to a pandemic), reduced mobility, or lack of motivation could be some of the reasons that older adults do not exercise. In this context, the care and health system for ageing populations may benefit from improved services through telerehabilitation that allows the monitoring of patients in their home environment—for example, in remote communities far from larger urban centers [39]. A promising option for simultaneous cognitive–motor training that lends itself to telerehabilitation is interactive game-based training, so called exergaming [40]. Exergames are any type of video game interactions that require the player to be physically active and move to play the game [41]. The rapid growth in new information and communication technologies has supported the development of several new virtual reality-based exergames for entertainment but also for gaming in rehabilitation settings or for disease prevention [42,43,44,45]. Several studies were able to demonstrate that exergame-based treatment is effective both in healthy and cognitively and physically impaired older adults [46,47] while also showing motivational benefits [48,49]. “Having fun while training” is assumed to have a positive impact on motivation, engagement, and compliance, thus influencing treatment effects [48,50,51]. Some games do not apply game design guidelines for older adults and are therefore not suitable for them [52]. Thus, it is important to consider the needs and constraints of the target population in order to provide individually tailored and enjoyable games [43,53,54].

VITAAL is an international project of the Active Assisted Living Programme (AAL), including multidisciplinary teams from different countries (Belgium, Portugal, Switzerland, and Canada), with the main goal of developing a new technology-based training solution that can be deployed at home for three target groups: older adults with mobility limitations, cognitive impairments, and urinary incontinence. The VITAAL exergame is developed to be finally used by autonomously living older adults at their homes because in-home interventions to prevent functional decline are often preferred by older adults [55,56]. Prior to conducting intervention studies, however, it is important to test the usability and acceptance of the newly developed solution by the target groups. The aim of this study was to investigate the usability of the VITAAL exergame in older adults with mobility limitations.

## 2. Materials and Methods

### 2.1. Study Design and Participants

To determine whether the VITAAL prototype can meet the usability needs of older adults with mobility limitations, a usability study was planned and designed according to a usability framework [57]. The study design consisted of a mixed-methods design where individual subjects were asked to try out the VITAAL exergame prototype during one exergame session. From February to September 2020, potential participants were recruited through local contact persons at Physio SPArtos (Interlaken, Bern, Switzerland) and public advertisements in local newspapers, in the surroundings of Interlaken (Bern, Switzerland). The investigation took place at a single measurement time point, including screening, a 30 min exergame session, and study measurements at Physio SPArtos. The ETH Zurich Ethics Committee (Zurich, Switzerland) granted ethical approval for the study (protocol number EK 2019-N-95). All participants were fully informed prior to participation and signed an informed consent form according to the Declaration of Helsinki before conducting any measurement.

A health questionnaire was completed to screen whether the potential participants were eligible to be included in this study. Additionally, two screening measurements were conducted: the Short Physical Performance Battery (SPPB; assessing physical functionality resp. mobility limitations) and the Montreal Cognitive Assessment (MoCA; assessing cognitive status). Participants fulfilling all the following inclusion criteria were eligible for participation in the study: (1) age ≥ 60 years, (2) living independently, in a residency dwelling, or with care, (3) being able to stand straight for minimal 10 min without aids, (4) visual acuity with correction sufficient to work on a TV screen, (5) SPPB ≤ 10 as a threshold value signifying functional impairment [58,59,60]. Participants exhibiting one of the following criteria were excluded from the study: (1) severe mobility impairments that prevent them from training participation, (2) severe cognitive impairments (below the 1^st^ percentile according to calculations of Thomann et al. [61]), (3) severe acute or uncontrolled health problems (e.g., recent cardiac infarction, uncontrolled high blood pressure or cardiovascular disease, uncontrolled diabetes), (4) orthopedic or neurological diseases that prevent them from training participation, (5) rapidly progressive or terminal illness, (6) chronic respiratory disease, (7) condition or therapy that weakens the immune system (e.g., autoimmune diseases), (8) cancer, and (9) serious obesity (BMI > 40 kg/m2). Hwang et al. defined a sample size rule for usability studies, stating that 10 ± 2 would be the optimal sample size [62]. Accordingly, a minimum sample size of 10–15 was defined for the intended study.

### 2.2. Exergame Intervention

The VITAAL exergame is a multicomponent exergame training system whose training content is focused on the prevention and slowing of physical and cognitive decline and its consequences. The training content consists of three major components: strength training, balance training, and cognitive training. For strength training, Tai Chi-inspired movements, which are a combination of classical strength exercises and Tai Chi movements, were included. Since Tai Chi is mainly performed in a semi-squat posture, it places a large load on the muscles of the lower extremities [63,64]. For balance training, step-based training was included in the VITAAL exergame, as the execution of rapid and well-directed steps has been shown to be effective in preventing falls [65,66,67]. Both Tai Chi-inspired exercises and step-based exercises, combined with challenging game tasks, provide a “holistic” physical activity requiring motor functions, cognition, and mental involvement [68]. Moreover, Tai Chi-inspired training, and step-based training could be more motivating and joyful than standard exercises. Some cognitive training is already included in these training components as they represent simultaneous cognitive–motor interaction and require motor and cognitive functions. However, because specific attentional and executive functions are important for walking abilities and safe gait [27,28,29,30,31], the VITAAL exergame explicitly targets these neuropsychological functions (selective attention, divided attention, inhibition/interference control, mental flexibility, and working memory).

The VITAAL exergame should be easy to use autonomously by older adults in their homes. As a web-based exergame, it is designed to run anywhere if there is a Bluetooth- and internet-enabled device with a screen (e.g., PC, laptop, tablet, etc.). The front end is designed for large screens and may ideally be visualized on a TV screen. The system is supported by a backend (main server supporting the whole service and data storage), a web portal with information about interventions, sessions, results, etc., and two wearable inertial measurement unit (IMU) sensors for measuring the stepping movements and game navigation. The IMUs incorporate a 32-bit Arm Cortex M4F processor (Nordic nRF52) and are equipped with a tri-axial gyroscope and a tri-axial accelerometer (Bosch BMI160). See [69] for further details. The two inertial sensors are placed on the shoes and sense accelerations and angular rotations caused by movement. The sensors communicate via Bluetooth with the software running on the web-enabled device. By communicating, the sensors provide information about the correctness of the movement, which is integrated into the game in real time, so that the participant receives immediate feedback.

From March to September 2020, single measurement appointments were arranged, including a gaming session with the VITAAL exergame as described above. Participants were asked to independently try to use the training system after a short introduction by the instructors. They had to complete 10 min training of each component (strength, balance, cognition). Figure 1 shows the different game themes, including the minigames for each exercise component, of which two minigames were played per component for roughly the same amount of time. The minigames were freely chosen by the participants. In the strength training, the movements of the exercises had to be imitated by the participants. These movements were not registered by the sensors; however, to ensure better understanding of the movements, frontal and side view of the exercises were provided. The participants did not receive any feedback during the strength exercises. For the cognitive and balance games, participants’ step movements were registered by the sensors on the top of their feet, so that auditory and visual feedback on correct and incorrect movements was immediately displayed on the screen. During the gaming session, a usability protocol was used to note observations and direct feedback to the exergame in a so-called think aloud method [70]. To avoid influencing the training session, the experimenter did not provide feedback to the participants during the game. After the training sessions, questionnaires and interviews were used to evaluate the usability and collect further feedback. The VITAAL system setup and the exergame session are illustrated in Figure 2.

### 2.3. Primary Outcomes

Quantitative and qualitative assessments were combined in a mixed-methods approach to obtain more complex answers to the research question. Similar designs have been used in previous studies evaluating the usability of exergames among older adults [49]. During the exergame session, a usability protocol was completed, followed by a questionnaire and a semi-structured interview after the session. The supervisor was a trained physical therapist experienced working with geriatric populations.

#### 2.3.1. Qualitative Observation (Usability Protocol)

A usability protocol was filled in by the supervisor during the exergame session, noting the participants’ feedback and the supervisor’s observations. During the intervention, participants were encouraged to use the “think aloud” method by bringing up anything that came to their mind while playing the exergame [71]. The usability protocol covered the following main categories: (1) VITAAL exergame interaction, (2) game design, (3) emotions, (4) exercises, and (5) risks/limitations.

#### 2.3.2. System Usability Scale

The System Usability Scale (SUS) [72] was applied after the exergame session to provide a global view of subjective assessments of usability. The questionnaire consists of ten items rated on a 5-point Likert scale (1 = “strongly disagree” to 5 = “strongly agree”). For the calculation, the score of each odd-numbered question was subtracted by one, while the score of each even-numbered question was subtracted by five. The total score was then multiplied by 2.5, resulting in a final score ranging from 1 to 100, similar to a percentage score. The SUS is a scientifically validated, reliable (e.g., ω  =  0.91 [73]), and easily applicable instrument [72,74], which has been frequently used in other exergame studies [49,75,76,77,78]. Based on the acceptability ranges of Bangor [79], we considered an SUS score of at least 70 to be an “acceptable” solution (adjective ratings: 52 = “ok”, 73 = “good”, 85 = “excellent”, 100 = “best imaginable”).

#### 2.3.3. Qualitative Interview (Semi-Structured Interview)

A semi-structured interview was conducted by the supervisor immediately after the exergame session. All interviews were audio-recorded and lasted between 7 and 14 min (mean 10.5 min), without taking any notes to provide a natural and uninterrupted conversation between the interviewer and the participant. The interviews took place in a quiet room at Physio SPArtos, ensuring that memories were still present and not distorted. Open and closed questions were asked about their exergame experiences in the following categories: (1) overall, (2) game, (3) VITAAL exergame/controller, (4) body and mind, (5) motivation, (6) training, (7) comparison to conventional therapy, and (8) suggestions. The guideline-based interview is provided in the Appendix A. Each interview was audio-recorded and fully transcribed in written form. The transcripts were not returned to the participants for revision and correction, as it was thought that post-processing after the passage of time might lead to bias.

### 2.4. Other Outcomes: Intensity Rating

Participants rated their physical and cognitive exertion after each minigame played during the exergame session on a scale from 0 to 10, anchoring the endpoints where 0 is “easy” and 10 is “difficult”. Targeting moderate to vigorous exercise intensity, which is the recommended intensity for older adults [80], ratings in the range of five and somewhat above were expected.

### 2.5. Data Analysis

SPSS 23.0 for Windows (SPSS Inc, Chicago, IL, USA) was used for statistical analysis of the quantitative data. Descriptive statistics were generated for participants’ characteristics, the SUS, and the intensity ratings. The usability protocol data were, as a first step, electronically recorded and tabulated in Microsoft Word. The transcribed version of the results was read several times by two of the authors (MT and LR) to gain a better understanding of the data. Subsequently, the following main categories were established: (1) VITAAL exergame interaction, (2) game design, (3) emotions, (4) exercises, and (5) risks/limitations. The data were then coded according to the categories to derive main statements. In a further step, the main statements for each category were divided according to positive and negative aspects. Observations and feedback were counted, avoiding multiple counting of the same statements from the same participant. For increasing the quality of the analysis procedure, coding and data analysis were performed and cross-checked by two of the authors (MT and LR). For the qualitative interview analysis, the audio data were transcribed in a written format in Microsoft Word after listening to them several times. Subsequently, the transcript was read repeatedly before it was processed following a qualitative content analysis according to Mayring et al. in Refs. [81,82] using the online software QCAmap [82,83,84]. Key questions from the guideline-based interviews were assigned to the appropriate content analytic procedures (i.e., inductive category formation or deductive category assignment). The inductive category formation questions were analyzed by establishing a selection criterion and level of abstraction. Deductive category assignment was chosen when the research questions allowed for the formulation of nominal or ordinal categories prior to processing the transcript. After category assignment was completed, the transcripts were analyzed and coded line by line, resulting in a list of categories. These lists were then divided into major categories. Interview responses were counted, avoiding multiple counting of the same statements from the same participant’s interview transcript.

## 3. Results

A total of 13 participants (9 female, 80.5 ± 4.9 years, range: 71–89) gave informed consent for the study, of whom all completed the measurement. A detailed overview of the participants’ demographic characteristics and screening measures is presented in Table 1. No adverse events occurred before, during, or after the exergame session.

### 3.1. Primary Outcome: Usability

#### 3.1.1. Usability Protocol

Participants’ main feedback towards the VITAAL exergame and further observations from supervisors during the exergame session are summarized in Table 2.

#### 3.1.2. System Usability Scale

The usability of the VITAAL exergame was rated with a mean SUS score of 58.3 ± 16.5 (min = 30, max = 85, *n* = 13), which represents an adjective rating between an “ok” and “good” acceptance level for usability [79]. The initially intended score of 70 was not reached. For the SUS, missing item ratings were replaced by the average response of the actual respondents (rounded to whole score numbers), to still be able to calculate the summary score [85]. In total, three questions were not scored and replaced. The usability ratings of the ten SUS items and an average rating over all items are summarized diagrammatically in Figure 3, where lighter shades of yellow indicate negative reactions and darker shades of yellow indicate positive reactions.

#### 3.1.3. Qualitative Interview Analysis

The result of the analysis of the interviews and the coding of the quotes was summarized in seven categories: (1) game, (2) VITAAL exergame interaction (3) body and mind, (4) motivation, (5) training, (6) comparison to conventional therapy, and (7) suggestions. For inductive questions, “*n*” describes the number of subjects who made this statement in the interview, while, for deductive questions, “*n*_total_” describes the number of subjects who answered this question.

##### Game

The most popular minigames, game design, comprehensibility, and game structure were analyzed under the heading of “games”.

***Popular Games***. Several fun games were named. The most popular minigames were “Mommy Chicken” (*n* = 5) and “Falling Books” (*n* = 4), which address balance, and “Healthy Food” (*n* = 2), which trains cognition. To enjoy the game play, respondents mentioned that it is important to understand the game (*n* = 5), that they need to be challenged (*n* = 3), or that they like to have a personal relation to the game content (*n* = 4).

*P09: The diet, with healthy food. Because healthy eating is important to me*.


*P10: Mommy Chicken. Because you could go up or left/right or down. Is there enough time or not? Will the chicken catch me or not? …*



*P13: The bookcase because it is very familiar to me from home.*


***Game Design.*** In general, the game design was positively described (*n* = 4). The visual design of the minigames was indicated to be good (*n* = 7), clear (*n* = 2), beautiful (*n* = 1), fun (*n* = 1), and interesting (*n* = 1).


*P06: The other minigames were very nice.*

*P10: … The game environment was good and interesting. You knew where you had to go. The picture is good for the game.*


Some participants criticized that the minigames could be designed a little more accurately (*n* = 2), more realistically (*n* = 2), and more vividly (*n* = 1). Furthermore, in the minigame “Healthy Food”, the food was not always recognized (*n* = 2).


*P13: The pictures were not so clear. I didn’t necessarily see a vegetable there…. Is that a vegetable or not? Maybe it shouldn’t be drawn so modern, but in a way that you can see more of what it actually is.*


Most participants did not really notice the music (*n* = 4), while another participant found the rhythm of the music helpful during the minigames (*n* = 1).


*P08: I didn’t notice any music, but I’m not that musical either.*



*P10: … Acoustics were good, the rhythm helps.*


Considering game feedback, some participants wished to see their own progress over time (*n* = 2), while others were satisfied and did not want any further feedback (*n* = 2).


*P04: I would like to know if I was any better or not, to know the progression.*



*P06: The feedback was perfect. I have always seen the success very well. I have not missed anything.*


***Game Structure***. Most participants considered that the game was structured in a comprehensible way (*n* = 9, 81.8%; *n*_total_ = 11).


*P06: Yes, the minigames were structured in an understandable way. I didn’t have to think much…*


##### VITAAL Exergame Interaction

The interaction and experiences with the VITAAL exergame were critically discussed with the interviewees, whereby the following four subcategories emerged: comprehensibility, game control, sensors, and experiences.

***Comprehensibility***. At the beginning, the handling of the exergame was perceived as somewhat difficult and unclear (*n* = 5) and required an explanation (*n* = 4). Afterwards, it was understood by most users (*n* = 6).


*P02: It took me a long time to understand what I had to do. It was difficult at the beginning, but not afterwards.*



*P06: … It needs someone to do an introduction on the screen, but afterwards it goes very well.*



*P08: … Explanations would have been good. One could not know what was coming.*


***Game Control***. In general, the game control was understood quickly or after some time (*n* = 6, 75.0%; *n*_total_ = 8). Controlling the game using the sensors was good (*n* = 1), direct, and not complex (*n* = 1).


*P12: Yes. I liked that the game controls were direct and not very complex.*


Some participants noticed that the sensors did not always react equally well (*n* = 6), which could lead to a feeling of uncertainty (*n* = 1) or impatience (*n* = 1) in participants.


*P03: Steps forward and backward were recognized less well compared to left and right. A step to the left was detected better than a step to the right.*



*P06: The step backwards into the middle was sometimes refused. This bothered me a little and made me feel insecure. I did not know if I was causing this.*



*P10: Sometimes I was a bit too impatient until the signal from the sensor reached the screen. The screen did not follow.*


***Sensors***. Some participants had difficulties recognizing the arrow on the sensors, which indicates the correct orientation on the feet (*n* = 2).


*P10: The arrow is hard to see if you have black shoes. You have to make sure that the arrow is facing forwards.*


Almost all participants found the sensors not easy to handle or could only use them with help (*n* = 12, 92.3%; *n*_total_ = 13).


*P06: Yes. The introduction on the screen was not very clear, I needed help… But otherwise, it is actually very simple.*



*P08: I did not find them easy to handle. I have never put on such sensors.*


The participants mentioned no or only single technical problems with the sensors (*n* = 3, 60.0%; *n*_total_ = 5) during the exergame session.


*P14: No, there were no technical problems. Once, the sensors did not react properly.*


***Experiences***. Overall, the exergame experience was positive and described as good (*n* =3), interesting (*n* = 2), and fun (*n* = 2).


*P12: Yes, it was fun. I haven’t done anything like that in a long time.*



*P15: I really liked the handling and experience of the VITAAL exergame, I would do it more often.*


##### Body and Mind

The topics flow, awareness of game performance, focus, movement awareness, and feelings have been summarized within this category.

***Flow***. The majority of the participants were absorbed or felt immersed in the game (*n* = 6).


*P10: Yes, I felt immersed in the game. That has to do with concentration as well. You are immediately focused.*


***Awareness of Game Performance.*** Roughly half of the participants said that they were always aware of their individual game performance (*n* = 7, 58%; *n*_total_ = 12), while almost as many participants were not aware of the feedback provided during and after the game (*n* = 5, 42%; *n*_total_ = 12).


*P10: Yes, I noticed how good or bad I was in the game. Especially with the cakes I was wrong for a while, there you had to concentrate a lot. I also saw the points and knew immediately whether I was good or bad.*



*P08: Did I receive points? I did not notice.*


***Focus***. Participants reported that they mostly focused on the cognitive aspect of the minigames (*n* = 5), whereas few subjects were more concerned about executing the steps correctly (*n* = 2).


*P06: I focused more on the cognitive, 100%.*



*P12: I focused on solving the task as determinedly as possible and not letting myself get distracted. I focused more on the game.*



*P15: Back and forth, front and back … I focused more on the steps and the body than on the game.*


Another interviewee mentioned that the focus was on the coordination of both physical and cognitive (*n* = 1).


*P14: I focused on both, physical and mental. You have to coordinate both.*


***Movement Awareness***. All participants experienced the movements as natural (*n* = 7, 100%; *n*_total_ = 7).


*P07: Yes, the movements felt natural to me.*


***Feelings***. Overall, the participants experienced positive feelings while playing the exergames. The subjects felt good (*n* = 6) and enthusiastic (*n* = 1). The minigames were described as fun (*n* = 4), captivating (*n* = 1), diverse (*n* = 1), and interesting (*n* = 1).


*P03: I felt good while playing. It was fun, varying, and enjoyable.*



*P06: … I am enthusiastic about Exergame, I think it is very good. … after 6–7 min you are part of the game.*


A few participants (*n* = 3) felt some uncertainty in the beginning of the exergame session.


*P12: … There were fluctuations at the beginning, a bit of uncertainty about what it’s all about, what I have to do. After that, it became a bit more relaxed.*


##### Motivation

In the interview, motivating and non-motivating factors were identified, and it was discussed whether the future use of the exergame is conceivable.

***Game variability***. Presenting a variety of minigames (*n* = 2) is motivating. Consequently, the exergame itself was perceived as encouraging movement by most participants (*n* = 6, 66.7%; *n*_total_ = 9).


*P10: It was interesting, especially that it had different minigames.*



*P15: Yes, it really motivated me. I had to move a lot.*


***Improvement***. Showing the players that their performance can still be improved is an additional motivational factor (*n* = 2).


*P09: The game is fun when you can learn something and improve your reaction.*


***Challenge***. For motivation to be maintained, the difficulty level should be adjusted (*n* = 1) so that the games remain challenging (*n* = 1) and one become ambitious (*n* = 1).


*P10: Yes, I could imagine that the game is still fun after playing it several times and that you become ambitious over time.*


***Understanding/Education***. A few participants (*n* = 2) mentioned the understanding and awareness of what they are exercising for as a motivator.


*P14: It’s certainly good for the body when you do something like this.*


***Exercise Variability.*** The subjects did not find it motivating when the exergame offered them too few or too monotonous physical exercises (*n* = 4).

***Game Design***. If the game design does not correspond to the individual’s wishes, it can have a demotivating effect on the subjects (*n* = 1).


*P13: I can’t do anything with this subject matter and these drawings. It could be that it becomes boring. For it not to be boring, the subjects and drawings would have to be more realistic, like photographs, not sketches.*


***Future Use***. Most participants think that the VITAAL exergame would still be fun after playing it several times (*n* = 8, 80.0%; *n*_total_ = 10) and could imagine using such games as part of therapy, in addition to the exercise that therapists traditionally offer (*n* = 6, 66.7%; *n*_total_ = 9). The majority of interviewees can also picture using the training at home or in a center for older adults (*n* = 8, 66.7%; *n*_total_ = 12).


*P10: Yes, I could imagine it, if you could download it. For the brain, coordination, agility… it has a little bit of everything in it.*


##### Training

In the interview, participants shared their training experiences and provided insights into training challenges, effort, concentration, training duration, and safety.

***Challenge***. The majority of participants experienced the exergame as challenging in terms of cognition (*n* = 5), balance (*n* = 2), physical effort (*n* = 1), coordination (*n* = 1), and correct movement of steps (*n* = 2).


*P03: Yes, it was challenging. Especially mentally.*



*P12: To really step on the ground with the tip of your foot was challenging.*


The squats were mentioned to be strenuous (*n* = 1), and the exhaustion was felt in the back (*n* = 1).


*P14: In the back was the effort, there was fatigue.*



*P15: … especially the squats at the end. That’s when I needed a rest.*



*In contrast, some subjects wished for a more physically demanding training program (n = 5).*



*P09: … More physically demanding would also be good.*


***Training effort.*** Most of the participants indicated that they did not feel optimally or only partially challenged (*n* = 4, 80%; *n*_total_ = 5), which was also reflected in their opinion about physical effort. Several participants stated that they did not or would rather not have to make any physical effort (*n* = 7, 70.0%; *n*_total_ = 10). The cognitive training was perceived as both demanding (*n* = 4, 57.2%; *n*_total_ = 7) and not demanding (*n* = 3, 42.8%; *n*_total_ = 7) to almost the same degree.

The majority of the interviewees believed that the desired functions, cognitive and physical, were being trained (*n* = 8, 88.9%; *n*_total_ = 9).

***Concentration***. Several subjects mentioned that they had to really concentrate during the training (*n* = 3).


*P15: You had to really concentrate, and make sure you took the right steps.*


***Training Duration.*** Most of the participants felt comfortable with a training time of 30 min (*n* = 5).


*P12: It should not be much more at a time, that half hour was good.*



*P14: That was good. In the beginning, the training should not be longer.*


***Safety***. In general, the participants felt safe and were not afraid of falling during the training (*n* = 5, 100%; *n*_total_ = 5).


*P09: No, I have never been afraid to fall.*


##### Comparison to Conventional Therapy/Exercises

When comparing the VITAAL exergame to conventional exercise therapy, different opinions were mentioned. On one hand, the VITAAL exergame was found to be more challenging (*n* = 2), cognitively more strenuous (*n* = 2), and physically more exhausting (*n* = 1).


*P15: In my head, I had to do more compared to the other therapies.*


On the other hand, participants experienced the VITAAL exergame as not exhausting (*n* = 2), easier (*n* = 1), cognitively less challenging (*n* = 1), and physically less strenuous (*n* = 1).


*P04: It is easier compared to therapy.*


A minority found the VITAAL exergame to be equally strenuous compared to conventional therapy or other activities (*n* = 1).


*P02: About the same effort compared to therapy/everyday life.*


##### Suggestions

Participants mentioned some ideas for adapting and improving the exergame. They expressed a wish for more intensive physical exercises (*n* = 3), more challenges (*n* = 1), more minigames (*n* = 1), and minigames with other animals (*n* = 2).


*P01: It just doesn’t have enough body intensity in it. The body is not used enough in this game.*



*P15: Yes, I would have more ideas, minigames with animals from the mountains such as chamois or ibex.*


#### 3.1.4. Other Outcome Results

The ratings of the physical and cognitive effort of the different minigames using a scale from 0 to 10 (0 = “easy” to 10 = “difficult”) are presented in Table 3.

## 4. Discussion

The aim of this study was to explore the usability of the VITAAL exergame prototype in older adults with mobility limitations. To the best of our knowledge, this is the first study investigating the usability of an exergame in older adults with mobility limitations, incorporating target end users, aged 60 years and older (mean age 79.6), in the initial stages of active prototype testing in a user-centered design approach. Feedback from users has previously been identified as important for the development and quality of innovative (tele-)rehabilitation approaches [86,87].

In this study, the SUS was used to quantitatively assess the usability of the VITAAL exergame prototype. The obtained score (58.3 ± 16.5) revealed a user-friendliness level between “ok” and “good”. This was below the initially intended score of 70, which would have been needed to present an acceptable exergame [79]. According to Bangor et al., a SUS score below 50 can be interpreted as having a non-acceptable system, whereas a score between 50 and 70 is in a marginal acceptability range [79]. The worst-rated items influencing the total score are in line with statements of the usability protocol and the interview, indicating that, for example, tightening and connecting the sensors or performing the calf raises would not have been possible without help in most participants. Furthermore, many participants wished for more instructions and explanations regarding the main board, the game control, and the minigames themselves. Interestingly, roughly the same number of participants mentioned that the use of the exergame was clear after some time, indicating that a short familiarization and learning period is required within the first training session. This is further supported by positively rated items: “*I found the system unnecessarily complex*” (77% disagreed), “*I would imagine that most people would learn to use this system very quickly*” (54% agreed), “*I found the system very cumbersome to use*” (77% disagreed). Older adults often have limited knowledge of technology; thus, ensuring a technology-based training system that can provide technical confidence (for example, by a simple setup, stable connections, and intuitive game environment) is very important [52]. Thereby, a better playing experience and more successful training can be achieved. Previous literature has highlighted the importance of age-appropriate design and impeccable technical functionality for the usability of exergames as well [76,88,89]. In other exergame studies, the SUS score was usually around 70 points or higher and they included participants with a mean age in the early seventies [75,77,90]. Compared to the mean age of the subjects in this study (80.5 ± 4.9 years), an age difference of almost one decade is evident. A study by Bangor et al. has shown that a significant correlation between age and SUS score exists, revealing that the age of the user might have some negative influence on the usability rating [91]. This was also supported by a study by Vaziri et al. investigating user experience and technology acceptance for a fall prevention system by analyzing the SUS considering the participants’ age [76]. The study showed an overall SUS score of 62. However, participants with an age younger than 72 years scored the system with 72 points and participants aged older than 72 years scored the system with 53 points. In the present study, many SUS items were rated as neutral, which could be interpreted as having no opinion about these items. One reason for this could be that the SUS was filled in after just one appointment, while, in many usability studies, SUS is applied only after several appointments [75,90], allowing participants to form more experiences and build a more consolidated opinion about the exergame.

Despite the marginal rating of the SUS score and the aforementioned difficulties at the beginning of the exergame use, qualitative analysis gave a good insight into the overall user experience and resulted in generally positive feedback. The minigames were structured in a comprehensible way. They were beautifully as well as interestingly designed. In particular, minigames that were designed with a theme to which the person had a personal relation, such as animals or books, were well-received. Apart from this, one minigame (Healthy Food) was criticized for having food icons that were difficult to recognize. When it comes to game design for older adults, it is important that the graphical user interface can be adjusted and well-defined game icons are used [52]. Nevertheless, the different minigames were mostly described as fun and made the participants laugh. Furthermore, the exergame encouraged participants to move. High motivation seems to be crucial for the success of exergame training and training interventions in general. When players are motivated, higher training compliance can be expected and this, in turn, might increase training success [92]. The most motivating factors seem to be high game variability, ongoing challenge, and awareness of the (health) goal of an exercise. In contrast, it can be demotivating when the sensors do not react properly, which leads to uncertainty, frustration, and impatience in some participants. It is also not motivating when the desired functions cannot be trained.

In general, the exergame training was perceived to be cognitively more challenging than physically. However, most of the participants did not feel optimally trained and described the minigames as not physically demanding. This might be explained by the fact that the participants are/were quite active. Approximately 60% (*n* = 8) engaged in physical activities such as going for a walk or gymnastics more than three times per week. Furthermore, the same baseline difficulty level was applied for the exergame session for all participants to gain a better understanding regarding the starting level of the minigames. When exercising more than once with the VITAAL exergame, the difficulty level of the minigames will be automatically adjusted, using a progression algorithm based on the performance of the subjects. In addition, some participants mentioned that they really had to concentrate during the game and others needed an extra break between the exercises. The results of the 0 to 10 scale rating of the perceived physical and cognitive exertion gives another impression. The cognitive exertion was rated the highest in balance and cognitive training, with a rating above five, whereas the physical exertion was rated the highest in strength and balance training, reflecting “moderate” intensity in most minigames. This can lead to the assumption that at least moderate training intensities, which are recommended for older adults [80], were achieved. However, this is only an average result and must be interpreted with caution. The discrepancy with the usability protocol and the interview might be explained by the high variability in the individual ratings. When the VITAAL exergame and conventional exercise therapy were compared, this discrepancy in intensity was again reflected, as some found the VITAAL exergame and others the conventional therapy more strenuous and challenging. Even though the training content might not have been perceived as challenging enough, most participants felt comfortable with a training duration of 30 min. Moreover, around half of the participants seemed to have reached a flow state, as indicated by the feeling of absorption and immersion in the game. This experience is also underlined by the fact that some participants did not notice the feedback given during the minigames, which might have been a result of total concentration when conducting the task. Flow describes a state that occurs when individuals are attentive and engaged in certain activities [93], and when the individuals’ skills are well-balanced with the challenges. Furthermore, the flow experience has been shown to encourage exergame play and thus further promote health [94].

In summary, the feedback on the exergame was positive and the participants felt safe using it. Moreover, most could envision using such an exergame, in addition to their usual therapy, at home or in a center for older adults. Participants felt safe and were not afraid of falling during the training.

### 4.1. Limitations

Some limitations need to be discussed. First, the usability of the VITAAL exergame was assessed within only one exercise session and within a living-lab setting, which might not allow enough time to familiarize participants with the exergame, nor does it reflect the actual home environment of the older adults. Initial insecurities that would disappear within the first training sessions could have negatively influenced the results. Second, a basic difficulty level of the exercises was adopted, which may have resulted in feelings of not being optimally challenged. Third, to maintain objectivity, the analysis was not conducted by the same person who conducted the exergame session and the interview. However, this could also lead to misinterpretation of the qualitative data, as non-verbal impressions might be lost. Therefore, these data should be interpreted with caution.

### 4.2. Implications for the VITAAL Exergame

Based on the results of this usability study, minor implications for improving the current VITAAL exergame prototype for older adults with mobility limitations are presented:Another movement for the “exit function” should be considered, as calf raises seem to be difficult for older adults with mobility limitations.Design aspects such as contrast and size are important when it comes to the usability of exergames for older adults. Therefore, the size or contrast of the arrow on the sensors should be adjusted to ensure easier handling. Furthermore, the design of the food in the “Healthy Food” minigame should be revised and presented in a more realistic design.Especially for older adults, explanations on how the exergame is installed (sensor connection), how the game control works, and what the game tasks includes are required.Even though the step detection algorithm works quite well, minor changes should be made to react faster to participant movements.

## 5. Conclusions

The VITAAL exergame prototype received positive feedback and can be considered usable for older adults with mobility limitations, considering improvements to the system in terms of design, instructions, and technical aspects targeted at raising the acceptance level for usability. After this first evaluation of the newly developed exergame prototype, the results warrant testing of the feasibility of the adapted multicomponent VITAAL exergame and assessment of its effects on physical and cognitive functions, in comparison with conventional training.

## Figures and Tables

**Figure 1 ijerph-18-13422-f001:**
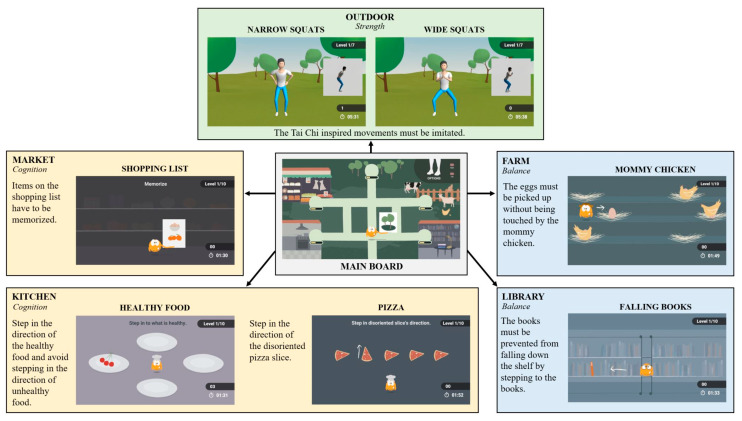
Overview of the VITAAL minigames. The VITAAL exergame focuses on three training components—strength, balance, and cognition—with the corresponding minigames based on different themes: outdoor, market, kitchen, farm, and library. During the strength training, the participants must imitate the movements simultaneously. For the cognitive games and the balance games, task instructions are given at the beginning of the game. These tasks can be solved with step-induced responses to the task, with immediate feedback on their correctness given during the games.

**Figure 2 ijerph-18-13422-f002:**
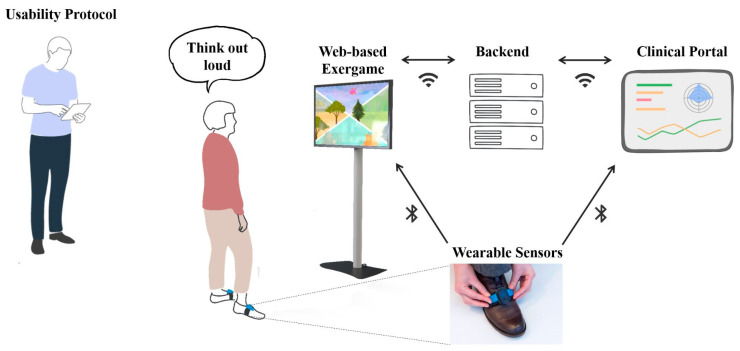
VITAAL system and exergame session setup. The VITAAL solution consists of a web-based exergame and a clinical portal. Both communicate with the backend via a wireless connection and with the sensors via Bluetooth. With the sensors placed on the feet, steps will be detected and real-time feedback within the game can be provided for cognitive and balance games. The participant is thinking out loud during the exergame session while the therapist observes and fills in the usability protocol.

**Figure 3 ijerph-18-13422-f003:**
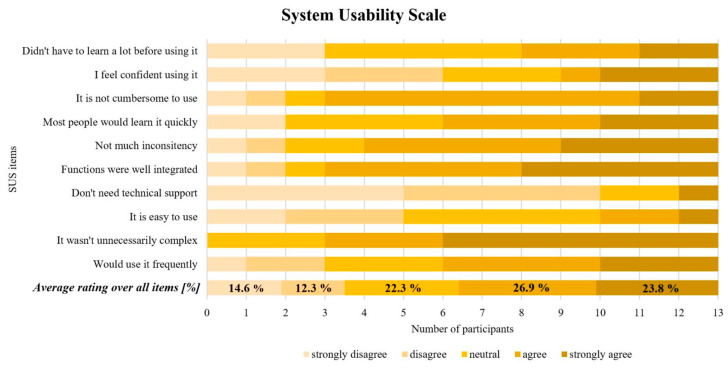
Summary diagram of the System Usability Scale. Ten SUS items and the average rating over all items including percentages.

**Table 1 ijerph-18-13422-t001:** Baseline demographic characteristics of participants and screening values.

Participant Characteristics	*n* = 13
Age in years	80.5 ± 4.9 (71–89)
Weight in kg	69.1 ± 13.6 (51–98)
Height in cm	164.3 ± 4.8 (158–172)
Education in years	11.9 ± 2.8 (9–19)
MoCA Score	26.9 ± 1.9 (24–30)
SPPB Score	8.5 ± 1.3 (6–10)
**Sex [*n*, %]**	
Female	9 (69.2)
Male	4 (30.8)
**Self-evaluation of muscle strength [*n*, %]**	
Very good	1 (7.7)
Good	2 (15.4)
Medium	7 (53.8)
Bad	3 (23.1)
I don’t know	0 (0.0)
**Problems with legs [*n*, %]**	
No	6 (46.2)
Sometimes	5 (38.5)
Always	2 (15.4)
I don’t know	0 (0.0)
**Fear of falling [*n*, %]**	
Never	7 (53.8)
Sometimes	4 (30.8)
Often	1 (7.7)
Always	1 (7.7)
**Number of falls during last 6 month * [*n*, %]**	
Never	8 (61.5)
Once	2 (15.4)
More than once	3 (23.1)
**Walking aids [*n*, %]**	
No	10 (76.9)
Cane/Stick/Crutch	3 (23.1)
Rollator	0 (0.0)
**Physical activity [*n*, %]**	
>3 x/week	8 (61.5)
1–3 x/week	5 (38.5)
1 x/week	0 (0.0)
No	0 (0.0)
**Use of video games in everyday life [*n*, %]**	
Yes	2 (15.4)
No	11 (84.6)
**Experience with exergames [*n*, %]**	
Yes	8 (61.5)
No	5 (38.5)
**Urinary incontinence * [*n*, %]**	
Yes	2 (15.4)
No	10 (76.9)
Missing	1 (7.7)
**Cognitive impairment * [*n*, %]**	
Yes	2 (15.4)
No	10 (76.9)
Missing	1 (7.7)

Data are mean values ± standard deviations (ranges) or number of participants per category (absolute and relative frequency). Montreal Cognitive Assessment (MoCA), Short Physical Performance Battery (SPPB). * Self-stated.

**Table 2 ijerph-18-13422-t002:** Summary of usability protocol with supervisors’ observations and participants’ feedback.

**VITAAL Exergame Interaction**	* **Summary:** * *Despite the explanations built into the game, additional guidance was needed on the technical equipment, the game controls, and the games.*	
	**Positive aspects**	**Negative aspects**
	The games are understood immediately (3) or after a short explanation (5) Game control by means of steps is well-understood after an explanation and some practice (6) The interaction with the exergame is interesting (6)	More detailed explanations of the start menu (5) and the game control by means of steps (5), especially their starting position in parallel stand (5), necessary Games need additional explanation (4); in particular, the “Pizza” minigame seems difficult (4) Step recognition was not always immediate (7); in particular, the recognition of the backward step seems to be inconsistent (3) Tightening the sensors is difficult and needs further explanation (11) Connecting the sensors with Bluetooth causes some difficulties (5)
**Game** **Design**	** *Summary:* ** *The games were found to be beautifully and interestingly designed. Nevertheless, recognizing specific objects was not always easy.*	
	**Positive aspects**	**Negative aspects**
	Beautiful (5), looks good (2), and is appealing in terms of design (1) Understandable (2), easy and clear to use (1) Interesting (design) (3) Good overview (main screen) (1)	Food icons not always very clear, so it was difficult to distinguish the healthy ones from the unhealthy ones (6) Calf raises icon unclear (3)
**Emotions**	** *Summary:* ** *The game is fun, makes you laugh, and motivates you to play. However, if the game is not successful, it can also lead to disappointment, frustration, and dissatisfaction.*	
	**Positive aspects**	**Negative aspects**
	The game makes you laugh (6), is fun (8) and motivating (4) The games are captivating (5) and exciting (1)	Frustration (4), annoyance (2), irritation (1), uncertainty (1) when the steps are not detected or detected incorrectly Disappointment (1), dissatisfaction (1), and annoyance (1) when making mistakes or not understanding the game
**Exercises**	** *Summary:* ** *The steps and the squats can be performed properly and correctly in most cases. Most of the games are not physically demanding, except for the squats. The games are mainly cognitively demanding and require concentration.*	
	**Positive aspects**	**Negative aspects**
	No additional breaks necessary (6) The exercises are taken seriously and performed with concentration (5) The exercises/steps are performed fast (3) and correctly (7) The training is physically demanding (3) and tiring (3) The step-based games were rated as cognitively hard (1), exhausting (4), and challenging (2); it also requires concentration (2) and a lot of thinking (2) The squats are physically demanding (4)	Additional breaks necessary (3) Some forget to go back to the starting position (2) Squats are not always performed well (3), because of fear from pain (1) The games are a bit slow (2), not very strenuous or challenging (2) The games were physically easy/not demanding (6) The games are not cognitively demanding/difficult (3) No upper body exercises (2) The movements are a bit boring (3) Problems with balance from time to time (2)
**Risks/** **Limitations**	** *Summary:* ** *The calf raises, which are needed as an exit function, are not possible for many or only possible with help. Knee pain can limit strength training.*	
	**Positive aspects**	**Negative aspects**
	Calf raises work well (3)	Calf raises not possible or only with help (6) Knee pain (4), which led to the termination of strength training for a few (2)

(*n*) = number of participants who made this statement or observations noted by the supervisor.

**Table 3 ijerph-18-13422-t003:** Physical and cognitive intensity of the VITAAL exergames.

	Physical (0–10 Scale)	Cognitive (0–10 Scale)
** *Balance* **		
**Falling Books [*n*]**	** *n* ** ** = 12**	** *n* ** ** = 9**
	5.7 ± 1.4 (4–8)	4.9 ± 2.8 (1–9)
**Mommy Chicken [*n*]**	** *n* ** ** = 8**	** *n* ** ** = 5**
	5.6 ± 1.7 (3–8)	5.6 ± 3.0 (2–9)
** *Cognition* **		
**Healthy Food [*n*]**	** *n* ** ** = 10**	** *n* ** ** = 8**
	2.5 ± 1.7 (0–6)	4.0 ± 2.2 (2–8)
**Pizza [*n*]**	** *n* ** ** = 10**	** *n* ** ** = 9**
	5.2 ± 2.5 (2–9)	5.9 ± 1.9 (3–8)
**Shopping List [*n*]**	** *n* ** ** = 3**	** *n* ** ** = 3**
	3.3 ± 4.2 (0–8)	5.7 ± 3.5 (2–9)
** *Strength* **		
**Narrow Squats [*n*]**	** *n* ** ** = 7**	** *n* ** ** = 7**
	6.6 ± 1.0 (5–8)	2.4 ± 2.2 (0–7)
**Wide Squats [*n*]**	** *n* ** ** = 3**	** *n* ** ** = 3**
	6.3 ± 2.1 (4–8)	2.0 ± 2.0 (0–4)

Data are mean values ± standard deviations (ranges) or number of participants per category; 0 is lowest and 10 is highest intensity on the 0–10 scale.

## Data Availability

The data presented in this study are available on request from the corresponding author.

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
