# Peer review of "Usability Study of a Multicomponent Exergame Training for Older Adults with Mobility Limitations"

_ijerph, 2021, doi:10.3390/ijerph182413422_

Round 1
Reviewer 1 Report
Ijerph-1459331
Title: Usability Study of a Multicomponent Exergame Training for Older Adults with Mobility Limitations
General Comments
The topic is interesting and well described in the manuscript.
The following comments should be addressed to better clarify some issues and to improve the readability of the manuscript.
Abstract – lines 31-33: role and meaning of this sentence in the manuscript is not clear. The same sentence is repeated at the end of the Conclusions.
Materials and Methods.
This section should be integrated with relevant information/clarification, at least with respect to the following issues: 1) mobility limitations: Authors state that the VITAAL exergame addresses older adults with mobility limitations, however the enrolled patients seem to be not fully characterized with respect to this relevant aspect: the exclusion criteria are clear, the inclusion criteria not so much and, as far as it is possible to understand from results, the set of enrolled volunteers seem trained (performing physical activity several times/week), in a good physical status, with no or poor need for walking aid, most of them experienced with exergames; 2) setting of the testing session: where are the patients examined? Is there an operator present in person at any time during the test? How do the volunteers communicate with operators/assistants?; 3) technical details on the sensors: type, technical features, accuracy, possible inaccuracies due to positioning, instructions for positioning (is it a self-positioning?); text at lines 160-162 is not clear.
Intensity Rating (par 2.4): why a scale with 11 points has been chosen, rather than 10?
Data analysis and statistics. Main concern regards the use of parametric descriptive statistics: was the normality of distributions checked? In similar analysis, non-parametric statistics is often necessary/preferred (in example, Tbel 3 reports mean and standard deviation over only 3 respondents). The other concern is about the replacement of missing data, as discussed in results paragraph 3.1.2: this topic is extremely relevant and the reason why missing data have been forced to average response (please consider the possible non-normal distribution of the results) should be well clarified. In any case, this is rather a topic to discuss in the Methods and highlighted again in the limitations.
General comments on the reported Results: the original responses from the volunteers, though interesting, might be moved from Results end included in the Supplementary material; in any case ,please check the Italics for all of the responses.
Table 3 caption is missing
Discussion.
The concerns raised above should be also discussed in this Section, especially in terms of possible limitations of the study. Among the others, if this Reviewer is right, usability has not been tested in a home environment, which should be well clarified as a possible limitation of the study.
Also, the use of the adjective “minor” for the implications of the usability test outcomes seems not adequate: some of the suggested adaptations and adjustments seem not to be so “minor”.
Conclusions should be reviewed accordingly.
Author Response
Abstract – lines 31-33: role and meaning of this sentence in the manuscript is not clear. The same sentence is repeated at the end of the Conclusions.
AUTHOR’S RESPONSE: We thank the reviewer for this comment. Feasibility studies are often performed before larger studies that study effectiveness of the intervention are implemented [see: Thabane, L., Ma, J., Chu, R. et al. A tutorial on pilot studies: the what, why and how. BMC Med Res Methodol 10, 1 (2010).]. Therefore we first tested the feasibility of our newly developed Exergame intervention. Based on the feasibility we can now consider performing larger scale trials in which we compare the effects of the new intervention against existing approaches.
We have revised the sentence to clarify this aspect and now write “The results warrant testing the feasibility of the adapted multicomponent VITAAL exergame and its effects on physical and cognitive functions, in comparison with conventional training, should be studied.” We also changed the sentence at the end of the Conclusions
Materials and Methods.
This section should be integrated with relevant information/clarification, at least with respect to the following issues: 1) mobility limitations: Authors state that the VITAAL exergame addresses older adults with mobility limitations, however the enrolled patients seem to be not fully characterized with respect to this relevant aspect: the exclusion criteria are clear, the inclusion criteria not so much and, as far as it is possible to understand from results, the set of enrolled volunteers seem trained (performing physical activity several times/week), in a good physical status, with no or poor need for walking aid, most of them experienced with exergames; 2) setting of the testing session: where are the patients examined? Is there an operator present in person at any time during the test? How do the volunteers communicate with operators/assistants?; 3) technical details on the sensors: type, technical features, accuracy, possible inaccuracies due to positioning, instructions for positioning (is it a self-positioning?); text at lines 160-162 is not clear.
AUTHOR’S RESPONSE:
Ad 1): Thank you very much for pointing this out. We defined mobility limitations based on a SPPB score of 10 or below. This is a commonly accepted threshold value for functional impairment. We added this information to the text together with references to clarify the rationale: “(5) SPPB £ 10 as a threshold value signifying functional impairment [56-58]”.
Ad 2): In lines 106-110 the information of the setting was given. We wrote, amongst others, in this section: “The investigation took part at a single measurement time point including screening, a 30 min exergame session, and study measurements at Physio SPArtos”.
Ad 3): Thank you for pointing this out. We have added the following information “The system is supported by a back end (main server supporting the whole service and data storage), a web portal with information about interventions, sessions, results, etc., and two wearable inertial measurement unit (IMU) sensors for measuring the stepping movements and game navigation. The IMUs incorporate a 32-bit Arm Cortex M4F processor (Nordic nRF52) and were equipped with a tri-axial gyroscope and a tri-axial accelerometer (Bosch BMI160). See for further details [62].”
Intensity Rating (par 2.4): why a scale with 11 points has been chosen, rather than 10?
AUTHOR’S RESPONSE: We have chosen this scale based on a World Health Organization guideline “Global recommendations on physical activity for health [https://www.who.int/publications/i/item/9789241599979]” where it is stated on page 16 that “On a scale relative to an individual’s personal capacity, moderate-intensity physical activity is usually a 5 or 6 on a scale of 0–10”. This scale is not scored through 11 points as can be seen from the figure in the PDF-file.
Data analysis and statistics. Main concern regards the use of parametric descriptive statistics: was the normality of distributions checked? In similar analysis, non-parametric statistics is often necessary/preferred (in example, Tbel 3 reports mean and standard deviation over only 3 respondents). The other concern is about the replacement of missing data, as discussed in results paragraph 3.1.2: this topic is extremely relevant and the reason why missing data have been forced to average response (please consider the possible non-normal distribution of the results) should be well clarified. In any case, this is rather a topic to discuss in the Methods and highlighted again in the limitations.
AUTHOR’S RESPONSE: We thank the reviewer for this important comment. However, we made it clear our study was a usability/feasibility study. A primary purpose of such studies is to assess the potential for successful implementation of the proposed main intervention studies and to reduce threats to the validity of these studies. In [Tickle-Degnen L. (2013). Nuts and bolts of conducting feasibility studies. The American journal of occupational therapy : official publication of the American Occupational Therapy Association, 67(2), 171–176. https://doi.org/10.5014/ajot.2013.006270] you can find a description of a typology we used to guide the aims of our usability/feasibility study we designed to support the later further development of randomized controlled trials. The article provides an example of the studies that are underlying the development of one rehabilitation trial. From this paper it becomes clear that the purpose of most usability/feasibility studies should be to describe information and evidence related to the successful implementation and validity of a planned main trial. Null hypothesis significance testing is not appropriate for these studies unless the sample size is properly powered. The primary tests of the intervention effectiveness hypotheses should occur in the main study, not in the studies that are serving as feasibility or pilot studies. Data from these studies can, however, be used to get an initial idea about needed sample sizes in such larger trials. For these reasons we reported mean and variance measures that are commonly used when parametric types of statistics are to be used.
General comments on the reported Results: the original responses from the volunteers, though interesting, might be moved from Results end included in the Supplementary material; in any case ,please check the Italics for all of the responses.
AUTHOR’S RESPONSE: Considering our reply above regarding the character of our study with a focus on usability/feasibility we think that especially the responses from the volunteers are important sources of information in the context of the study design we used. For this reason, we rather prefer to keep this information in the main document.
We have checked (and corrected) the Italics for the responses.
Table 3 caption is missing
AUTHOR’S RESPONSE: A caption has been added to Table 3.
Discussion.
The concerns raised above should be also discussed in this Section, especially in terms of possible limitations of the study. Among the others, if this Reviewer is right, usability has not been tested in a home environment, which should be well clarified as a possible limitation of the study.
AUTHOR’S RESPONSE: Based on the reviewer’s feedback we have added the limitation of not training in a home environment as an additional limitation.
Also, the use of the adjective “minor” for the implications of the usability test outcomes seems not adequate: some of the suggested adaptations and adjustments seem not to be so “minor”.
AUTHOR’S RESPONSE: We fully agree with the reviewer that the adjective *minor” is difficult to interpret or quantify. We deleted this word from this section of the manuscript for improved clarity.
Conclusions should be reviewed accordingly.
AUTHOR’S RESPONSE: Based on the reviewer comments we have reviewed and modified our conclusions.

Reviewer 2 Report
Revision of the manuscript:
Usability Study of a Multicomponent Exergame Training for Older Adults with Mobility Limitations
The authors have written a manuscript examining the usability of the VITAAL exergame in older adults with mobility limitations.
The authors indicated that Mobility is by far one of the most important factors affecting independence and quality of life in older adults. The study topic is interesting and important, according to the fact that 23 to 47% of older adults have mobility limitations which are potentially associated with higher risk of falls, disability, mortality and worsening of psychosocial health due to social isolation. However, the manuscript contains some errors which must be improved before publication.
Introduction:
Line 66: Please add references.
General comment: In my opinion Authors should emphasize the novelty of their work.
Materials and Methods
Line 142: Please add references regarding Tai Chi.
Line 222: Please provide the reliability of the SUS.
Results:
The results are illegible. They should be presented in a clearer form, e.g. in a table, or only the most important results should be presented.
Discussion:
Line 610: There is no need to repeat the results in the Discussion section.
General comment: In this section the Authors should refer to the studies on assessing virtual reality-based training.
Conclusion:
The result was below the initially intended score of 70, which would have been needed to present an acceptable exergame. The Authors should mention about that in the conclusion.

Author Response
Introduction:
Line 66: Please add references.
AUTHOR’S RESPONSE: References have been added.
General comment: In my opinion Authors should emphasize the novelty of their work.
AUTHOR’S RESPONSE: We thank the reviewer for this important comment. We underscored the novelty of our work in the last paragraph of the Introduction where we write “VITAAL is an international project of the Active Assisted Living Programme (AAL) including multidisciplinary teams from different countries (Belgium, Portugal, Switzerland, and Canada) with the main goal of developing a new technology-based training solution that can be deployed at home for three target groups: older adults with mobility limitations, cognitive impairments, and urinary incontinence. The VITAAL exergame is developed to be finally used by autonomously living older adults at their homes because in-home interventions to prevent functional decline are often preferred by older adults [55, 56].”
Materials and Methods
Line 142: Please add references regarding Tai Chi.
AUTHOR’S RESPONSE: References have been added.
Line 222: Please provide the reliability of the SUS.
AUTHOR’S RESPONSE: Information about SUS reliability has been added.
Results:
The results are illegible. They should be presented in a clearer form, e.g. in a table, or only the most important results should be presented.
AUTHOR’S RESPONSE: We thank the reviewer for this important comment. However, we made it clear our study was a usability/feasibility study. A primary purpose of such studies is to assess the potential for successful implementation of the proposed main intervention studies and to reduce threats to the validity of these studies. In [Tickle-Degnen L. (2013). Nuts and bolts of conducting feasibility studies. The American journal of occupational therapy : official publication of the American Occupational Therapy Association, 67(2), 171–176. https://doi.org/10.5014/ajot.2013.006270] you can find a description of a typology we used to guide the aims of our usability/feasibility study we designed to support the later further development of randomized controlled trials. The article provides an example of the studies that are underlying the development of one rehabilitation trial. From this paper it becomes clear that the purpose of most usability/feasibility studies should be to describe information and evidence related to the successful implementation and validity of a planned main trial. For these reasons we reported the results as they are because these are the most important outcomes for the type of study design we used.
Discussion:
Line 610: There is no need to repeat the results in the Discussion section.
AUTHOR’S RESPONSE: We thank the reviewer for pointing this out. We deleted the repetition of the results.
General comment: In this section the Authors should refer to the studies on assessing virtual reality-based training.
AUTHOR’S RESPONSE: We are grateful to the reviewer for this remark. We have added a sentence to the first paragraph of the Discussion section pointing out the importance of user integration into the development of novel rehabilitation approaches by adding “Feedback from users has previously been identified as important for the development and quality of innovative (tele-)rehabilitation approaches [86, 87].”
Conclusion:
The result was below the initially intended score of 70, which would have been needed to present an acceptable exergame. The Authors should mention about that in the conclusion.
AUTHOR’S RESPONSE: We thank the reviewer for this suggestion and followed up on this comment by adding the necessity of raising acceptance level for usability in a next iteration of system development.
Reviewer 3 Report
The manuscript reports a usability study conducted on VITAAL, an exergame multicomponent training approach designed for older adults with mobility problems. In the study 13 older adults tested the newly developed exergame in a 30-min session. The authors concluded that minor improvements are required.
The subject is interesting. The Introduction and the methods used in the study are clearly described. The users play the exergame directed to train strength, balance, and cognition just a 30-minute session. A mixed method was used to combine quantitative and qualitative assessments, which it is appropriate for this type of studies. Please, clarify the use of SPSS 23.0. What statistical analyses were conducted with the program?
It is not clear that the 13 older adults that participated in this usability study had mobility limitations. This point requires clarification. It would be important to conduct the same usability study with participants that really had mobility limitations and were not physically active. Moreover, several training sessions and more participants would allow a better assessment of the system. Could the authors anticipate how many training sessions would be required to appreciate physical and cognitive improvements in the trainees?
Author Response
The subject is interesting. The Introduction and the methods used in the study are clearly described. The users play the exergame directed to train strength, balance, and cognition just a 30-minute session. A mixed method was used to combine quantitative and qualitative assessments, which it is appropriate for this type of studies. Please, clarify the use of SPSS 23.0. What statistical analyses were conducted with the program?
AUTHOR’S RESPONSE: SPSS 23.0 for Windows (SPSS Inc, Chicago, IL, United States) was used for statistical analysis of the quantitative data. Descriptive statistics were generated for participants’ characteristics, the SUS, and the intensity ratings using the SPSS program.
It is not clear that the 13 older adults that participated in this usability study had mobility limitations. This point requires clarification. It would be important to conduct the same usability study with participants that really had mobility limitations and were not physically active. Moreover, several training sessions and more participants would allow a better assessment of the system. Could the authors anticipate how many training sessions would be required to appreciate physical and cognitive improvements in the trainees?
AUTHOR’S RESPONSE: Thank you very much for pointing this out. We defined mobility limitations based on a SPPB score of 10 or below. This is a commonly accepted threshold value for functional impairment. We added this information to the text together with references to clarify the rationale: “(5) SPPB £ 10 as a threshold value signifying functional impairment [56-58]”.
It would be preliminary to anticipate the amount of training sessions from the data of our study. However, several guidelines give indications about the amount of training needed to achieve improvements in physical and cognitive functions. Further iterations in the development of this novel training approach would preferably orient on these guidelines for the determination of exergame training time and length. In line 66 of the introduction, we added the following references “Izquierdo, M., et al., International Exercise Recommendations in Older Adults (ICFSR): Expert Consensus Guidelines. J Nutr Health Aging, 2021. 25(7): p. 824-853. & Bangsbo, J., et al., Copenhagen Consensus statement 2019: physical activity and ageing. Br J Sports Med, 2019. 53(14): p. 856-858.” Both point out the necessary content of individualized exercise interventions to slow down disability progression in older adults before it impacts their quality of life.
Round 2
Reviewer 2 Report
Thank you for taking into account my comments.